# Dialogue between VE-Cadherin and Sphingosine 1 Phosphate Receptor1 (S1PR1) for Protecting Endothelial Functions

**DOI:** 10.3390/ijms24044018

**Published:** 2023-02-16

**Authors:** Olivia Garnier, Isabelle Vilgrain

**Affiliations:** University Grenoble Alpes, INSERM U13, CEA, BGE-Biomics, 38000 Grenoble, France

**Keywords:** endothelium, endothelial cells, adherens junctions, VE-cadherin, lysophopholipids, sphingosine-1-phosphate (S1P), sphingosine 1 phosphate receptor1 (S1PR1)

## Abstract

The endothelial cells (EC) of established blood vessels in adults remain extraordinarily quiescent in the sense that they are not actively proliferating, but they fulfill the necessary role to control the permeability of their monolayer that lines the interior of blood vessels. The cell–cell junctions between ECs in the endothelium comprise tight junctions and adherens homotypic junctions, which are ubiquitous along the vascular tree. Adherens junctions are adhesive intercellular contacts that are crucial for the organization of the EC monolayer and its maintenance and regulation of normal microvascular function. The molecular components and underlying signaling pathways that control the association of adherens junctions have been described in the last few years. In contrast, the role that dysfunction of these adherens junctions has in contributing to human vascular disease remains an important open issue. Sphingosine-1-phosphate (S1P) is a bioactive sphingolipid mediator found at high concentrations in blood which has important roles in the control of the vascular permeability, cell recruitment, and clotting that follow inflammatory processes. This role of S1P is achieved through a signaling pathway mediated through a family of G protein-coupled receptors designated as S1PR1. This review highlights novel evidence for a direct linkage between S1PR1 signaling and the mediation of EC cohesive properties that are controlled by VE-cadherin.

## 1. Introduction

The endothelium comprises a monolayer of endothelial cells (ECs) lining all vessels and functions as an important regulator of vascular homeostasis. It has important physiological properties that include mediating the balance between vasodilation and vasoconstriction, as well as the control of thrombogenesis and fibrinolysis. In addition, the endothelium provides a semipermeable barrier between blood and the interstitial space, which controls leukocyte trafficking and tumor metastasis. It is, therefore, important to understand the mechanisms that are involved in the maintenance of its functional integrity.

## 2. Adherens Junctions

The integrity of the endothelium depends on the cohesive strength between the ECs [1]. Several proteins exhibiting adhesive properties are involved in inter-EC contacts. These EC contacts, called adherens junctions, are present along the blood vessels. The interactions of cadherins with electron-dense complexes delineate the adhesive junctions that contribute to tissue integrity (Figure 1).

These specialized structures maintain the strength between ECs that is controlled by contractility and EC adhesive interactions. In adherens junctions, the transmembrane protein responsible for cell-to-cell adhesion is vascular endothelial (VE)-cadherin, which is cell-specific. It is present at all adherens junctions, and its Ca^2+^-dependent homotypic interactions are crucial for endothelial assembly. Thus, this protein is frequently used for labeling ECs at the early stages of vasculogenesis and angiogenesis [1,2,3]. Mouse embryos carrying a homozygous null mutation of the *VE-cadherin gene* died after 9.5 to 10 days due to a lack of vascular remodeling and angiogenesis [4]. This dramatic phenotype was similar to that obtained with partial truncation of the VE-cadherin cytoplasmic tail [5]. The homophilic adhesive interactions are mediated through the extracellular domain of the protein (EC1–EC5). The cytoplasmic domain which is associated with actin via an interaction of the C-terminal domain with β-catenin/α-catenin or plakoglobin/α-catenin is mostly responsible for the strength of the EC junction. In addition, association of VE-cadherin with vimentin through plakoglobin/desmoplakin stabilizes the anchorage of adherens junctions to the actin cytoskeleton [6]. VE-cadherin is involved in several biological processes, including endothelial cell migration, survival [5], contact-induced growth inhibition [1], vascular integrity [6], and the endothelial network [7]. Both the extracellular domain and the cytoplasmic tail of VE-cadherin are important for the maintenance of normal vascular integrity. The discovery of a monoclonal antibody directed against an extracellular epitope that is only accessible in ECs forming new angiogenic networks has allowed specific targeting of VE-cadherin in the tumor vasculature, despite ubiquitous VE-cadherin expression in normal vessels [8,9,10,11,12]. Importantly, targeting with the monoclonal antibody has demonstrated the major role of the extracellular domain of VE-cadherin [13]. Altogether, these data highlight the major role of VE-cadherin in the maintenance of EC junctions and the vascular barrier.

## 3. Tyrosine Phosphorylation of VE-Cadherin Has a Significant Impact on Barrier Dysfunction

Tyrosine phosphorylation of the cadherin–catenin complex is a potent mechanism that regulates the stability of cell–cell junctions. Multiple tyrosine kinases [14,15,16,17,18,19,20] and phosphatases [21,22,23,24,25,26,27] co-distribute with adherens junction components. Several factors, such as VEGF, TNF-α, platelet activating factor, thrombin, and histamine, are known to increase endothelial permeability by inducing the tyrosine phosphorylation of VE-cadherin [13]. The binding of VEGF to its receptor, VEGFR2, activates several autophosphorylation sites, including Y^951^ [28]. When phosphorylated, this provides a binding site for the T-cell-specific adaptor protein TSAd, which is a c-Src-homology-2 (SH2) domain-containing protein which regulates the activation of c-Src [29]. Among its different substrates in EC, c-Src targets VE-cadherin during vascular remodeling and angiogenesis [30,31]. Our laboratory has identified that the site 685 (Y^685^) in the cytoplasmic domain of VE-cadherin becomes phosphorylated in response to VEGF in vitro [30] and in vivo in mice [32,33], as well as in tumoral vasculature [34]. Other studies of microvasculature have reported that the phosphorylation of sites Y^658^ and Y^731^ following VEGF challenge occurs through reactive oxygen species and Rac1 [35]. Thus, several signaling pathways are involved in VEGF-mediated VE-cadherin phosphorylation, depending on the type of vessels and cells. In the case of TNF-α, it was shown that VE-cadherin was tyrosine-phosphorylated at sites Y^658^ and Y^731^ through the PI3K/P110alpha, Pyk2, and Rac1/Tiam1 cascades [36]. These phosphorylation sites (Y^658^ and Y^731^) are responsible for the uncoupling of β-catenin and p120, which are partners linked to actin cytoskeleton [37]. Although VE-cadherin tyrosine phosphorylation is barely detectable in the adult quiescent endothelium, it is a process required in several diseases driven by angiogenesis, including cancer, atherosclerosis, diabetic retinopathy, and arthritis.

## 4. Endothelium and Sphingosine-1-Phosphate (S1P)

**S1P**: sphingosine-1-phosphate (S1P) is a sphingolipid that comprises a class of lysophospholipid mediators that are important bioactive lipids. S1P originates from the platelets, and it has been shown to have a major role in a variety of biologic processes, which include DNA synthesis, neurite retraction, stress fiber formation, and membrane depolarization [38]. Many effects of S1P on vasculature are due to expression of the membrane receptor S1PR1 by the EC. Blood flow delivers S1P in micromolar concentrations, whereas tissue concentrations of S1P are usually in the nanomolar range [39]. In plasma and serum, S1P is distributed mainly in the high-density lipoprotein (HDL) fraction. In tissue, S1P is synthesized intracellularly by the actions of sphingomyelinase, ceramidase, and sphingosine kinase [40,41]. It has been demonstrated that S1P acts through specific binding to Edg (endothelial differentiation gene), the first receptor discovered during a search for immediate early genes that regulate endothelial cell differentiation. The ligand and its receptor were found to have some angiogenic properties. Indeed, treatment of ECs with S1P for 30 min resulted in a dramatic increase in actin stress fibers and cortical actin structures. Moreover, the activation of S1P receptors triggers several effects in EC, including proliferation, survival, migration, morphogenesis, adhesion molecule expression, and cytoskeletal changes [42,43]. Thus, S1P induced the proliferation of new vasculature, but stabilized the established vasculature. This is in sharp contrast to other angiogenic factors such as VEGF.

**S1P Receptors:** The characterization of the S1P receptors has provided new insight into this new angiogenic factor. S1PR1 is part of a family of seven-transmembrane-domain G-protein coupled receptors (GPCRs) [44]. The *S1PR1* transcript was cloned as an immediate-early gene induced during the formation of capillary-like networks by human ECs. Three putative, but less well-characterized, S1P-GPCRs have been identified [45,46]. Disruption of the *S1PR1* gene in mice caused embryonic lethality due to massive hemorrhage between embryonic days E12.5 to E14.5. The embryos underwent normal vasculogenesis and angiogenesis, but had a defect in the recruitment of mural cells to the vessel walls [47]. S1P has a high affinity (Kd 8nM) for its receptor, and the binding induces cytoskeletal rearrangement and subsequent endothelial barrier regulation. Lee et al. had previously shown that S1P up-regulates P-cadherin based adherens junctions in S1PR1 S1PR1-transfected HEK-293, then extended those investigations to demonstrate that the localization of VE-cadherin, α, β-, and γ-catenin at cell–cell junctions was dramatically increased within 1 h of S1P treatment [47]. That concept showing that the adherens junction assembly was under the dynamic control of S1PR1 in EC suggested that S1P might control EC integrity and functionality. From experiments using an S1PR1-specific agonist and antagonist in vivo, ex vivo, and in vitro, Gaengel et al. proposed that S1PR1 signaling regulates cellular adhesion, motility, and VE-cadherin localization. Moreover, an endothelial knockout of the *S1P1* gene led to damage to the adherens junctions and destabilization of VE-cadherin [48]. Altogether, the authors concluded that S1P/S1PR1 could protect the neovessels from hyperpermeability.

## 5. S1PR1 Signaling Pathways Controlling Endothelial Junctions

Although most of the overall physiological effects of S1P have been described, the details of the underlying signaling pathways remain only partially described. As for other GPCRs, S1PR1 is localized on the plasma membrane of EC. When it binds its ligand, S1PR1 is rapidly desensitized until it is re-cycled to the membrane. This process is also known for other GPCRs that are regulated by phosphorylation on serine residues of the C-terminus domain to uncouple GPCR/G-protein signaling. This uncoupling is followed by the recruitment of β-arrestin, which facilitates endocytosis, dephosphorylation, and resensitization [49]. Importantly, S1PR1 contains three tyrosine residues, including the tyrosine 143 (Y^143^) that mutational analysis has identified to be within the ERY motif between transmembrane domain III and the intracellular loop. The phosphorylation of this site has been shown to play a critical role in modulating the cell surface expression of S1PR1 [50]. S1PR1 phosphorylation at tyrosine Y^143^ was induced by the membrane-associated non-receptor tyrosine kinase c-Src. Another group has recently shown that Y^143^ phosphorylation of S1PR1 drives the receptor into the endoplasmic reticulum, which is dependent on the chaperone-binding immunoglobulin protein (BiP). BiP is also considered to be involved in protein folding and assembly, Ca^2+^ homeostasis, and regulation of endoplasmic reticulum stress signaling. Intracellular BiP was shown to be an important mediator of lung vascular injury through its ability to mediate EC permeability by virtue of activating Ca^2+^ signaling and promoting VE-cadherin disassembly [51]. BiP contains a nucleotide and a substrate-binding domain. The nucleotide-binding domain binds and hydrolyzes ATP. The activity of BiP is regulated by its allosteric ATPase cycle. Co-immunoprecipitation studies have shown that phosphorylated S1PR1 interacted with BiP through its ATPase domain. Furthermore, EC expressing Y^143^D-S1PR, which mimics the phosphorylated form of S1PR1, exhibited increased BiP-ATPase activity. Thus, the authors concluded that S1P induced the phosphorylation of S1PR1 at site Y^143^ and promoted the translocation of BiP to the cytoplasm. This led to increased BiP-ATPase activity, thereby augmenting its chaperone function. Altogether, these results show that BiP binds to endocytosed S1PR1 in the cytoplasm and promotes its transport to the endoplasmic reticulum. This identifies a new role for tyrosine phosphorylation of S1PR1, which supports its localization to a specific organelle and disruption of the EC barrier [52].

Mechanistically, S1PR1 was described to signal exclusively through the heterotrimeric G-protein composed of α, β, and γ subunits [49]. Gαs deletion in EC mimicked S1PR1 gene knockout, which caused severe hemorrhage and embryonic lethality around day E11.5. No modification of the mural cell coverage was seen, but a reduced expression of VE-cadherin at the EC junctions in embryos was observed. It seems that Gαs links the membrane-bound S1PR1 to its intracellular effector VE-cadherin, and thus controls vascular integrity [53].

S1PR1 can also activate the G-protein Gαi, which, in turn, induces mobilization of intracellular Ca^2+^ and thus leads to increases in the small GTPases Rac1 and Cdc42 [54]. As a consequence, activated Rac1 and Cdc42 stabilize cadherin–catenin complexes and lead to rigid cell–cell adhesion that regulates adherens junction assembly.

The PI3K pathway is also involved in S1PR signaling pathway by activating PDK1 (3-phosphoinositide-dependent protein kinase-1) and AKT. The pleckstrin homology domains of both PDK1 and AKT can bind to PIP3; thus, PDK1 phosphorylates and activates AKT at the amino acid threonine T^308^ in its catalytic site. AKT-1 then modulates Rac1-GTP binding through phosphorylation. The guanine nucleotide exchange factor specific for Rac Tiam1, T lymphoma invasion and metastasis gene 1, translocates to the cell–cell contact area. Then, Rac-1 activity recruits cortactin to the actin-nucleation complex, which allows the polymerization of actin filaments at the cell periphery that distend the membrane and lead to the spreading of ECs. Thus, the S1P challenge induced a dramatic increase in actin stress fibers in ECs, as well as an increase in the localization of VE-cadherin into discontinuous structures at cell–cell contact regions. S1P acts through another mechanism related to cell–matrix interactions via focal adhesions. S1P was shown to activate the cytoplasmic c-Src kinases in EC [55,56] and the focal adhesion kinases (FAKs) located at focal adhesions [57]. Because one of the primary functions of phosphotyrosine residues is to provide docking sites for proteins that contain c-Src-homology-2 (SH2) domains, the formation of protein complexes essential for the assembly of focal adhesion structures is facilitated. Among the different sites of phosphorylation that are found in the FAK structure, the amino acid tyrosine 397 (Y^397^) is the autophosphorylation site stimulated by cell–matrix adhesion. A result of this conformational change is the c-Src association through the c-Src-SH2 domain. In addition, there is a proline-rich sequence in FAK which acts as a high-affinity binding site for the SH3 domain of c-Src. The activation of c-Src leads to phosphorylation of FAK at several potential remaining sites, namely Y^407^, Y^576^, Y^577^, Y^861^, and Y^925^ [58]. These additional phosphorylation sites of FAK can affect actin fiber assembly and focal adhesion formation and turnover. Thus, c-Src family kinases in focal adhesion complexes affect not only cellular migration, but also EC shape and vascular permeability. Upon S1P challenge, the specific phosphorylation of FAK at the Y^576^ site plays an important role in S1P-induced focal adhesion formation, EC barrier enhancement, and EC chemotactic motility [59,60,61]. Conversely, upon EC cell–cell adhesion, it was shown that FAK directly phosphorylates the VE-cadherin cytoplasmic domain at tyrosine residue Y^658^ [62]. As FAK controls VE-cadherin-Y^658^ phosphorylation in response to VEGF and TNF, these data support the importance of FAK activity in the regulation of cell–cell adherens junctions in addition to the canonical role of FAK in controlling cell–matrix focal adhesion dynamics. VE-cadherin-Y^658^ is located within the p120-catenin binding region, and mutagenesis results support a model whereby Y^658^ phosphorylation disrupts p120-catenin binding and may induce site-selective modulation of VE-cadherin, thus affecting adherens junction stability [37]. Taken together, these data support a key role of S1P/S1PR1 in the regulation of FAK in the barrier function.

To further demonstrate the effect of S1P on VE-cadherin, Wang et al. [63] showed that the analog of S1P, (S)-FTY720-phosphonate((3S)-3-(amino)-3-(hydroxymethyl)-5-(4′octylphenyl)-pentylphosphonic acid) (Tys), induced both redistribution of the VE-cadherin complex to cell–cell junctions by binding with β-catenin, and, in association with cortical actin ring formation, anchored them to the cytoskeleton, in addition to improving cell–cell and cell–matrix interactions. The effect of Tys increasing barrier function in vitro was supported by measurements of increased transendothelial electrical resistance, which was significantly attenuated by a VE-cadherin functional blocking antibody. The marked increase in vascular permeability induced by the extracellular administration of anti-VE-cadherin antibodies demonstrates that VE-cadherin complexes are targets for Tys that are able to participate in EC barrier enhancement [13]. However, the stable analog of S1P exhibits some differences in effects on EC when compared to those of the native S1P. Nevertheless, the barrier regulatory effects of Tys reveal additional EC signaling pathways for preserving or reconstituting the vascular barrier that provides potential therapeutic utility.

Those effects of S1P on VE-cadherin have been further elucidated by experiments using mice and tissue cultures. An extensive post-transcriptional regulation of VE-cadherin by S1P was reported, including intracellular complex formation and trafficking, as well as membrane localization. The knockout of S1PR1 in EC induced a loss of VE-cadherin and VEGFR2 from the cell junctions in post-natal EC. The loss of S1PR1 in the mice embryos was associated with an endothelial hyperplasia and abnormal microvasculature in the aorta [48]. S1PR1 is known to regulate EC integrity and behavior by modulating junction protein localization and cell contractility [64]. S1PR1 has been shown, in vivo and in vitro, to increase junctional localization of VE-cadherin, thus increasing vascular stability and inhibiting VEGFR2 signaling. Indeed, under S1P stimulation, VE-cadherin co-immunoprecipitated with β-catenin, α-catenin, and γ-catenin. Conversely, the antagonist of S1PR1 increased cell sprouting, characterized by longer and more branched sprouts. Moreover, the deletion of S1PR1 and VE-cadherin genes was responsible for hyper-sprouting similar to that observed with VEGF. Thus, the collaboration between S1PR1 and VE-cadherin is important to inhibit cell sprouting and to maintain vascular integrity. More recently, the observation of VE-cadherin-mediated anastomosis of brain capillaries during early stages of brain vascularization demonstrated that the canonical Wnt-β-catenin signaling pathway could be counteracted by the S1PR1 pathway. Conversely, at later stages of vascularization, S1PR1 regulates VE-cadherin localization at the membrane and allows the maturation of the blood–brain barrier. In brain capillaries, Wnt signaling decreases to a low baseline level after lumen formation. In lumenized vessels, activation of S1PR1 signaling would occur, which would enable functional S1PR1 signaling for blood–brain barrier maintenance. As S1PR1 activated Rac1, the authors speculated that Wnt signaling may regulate Rac1-mediated S1PR1 signaling and thereby modulate junction localization of adhesion molecules. These data indicate that S1PR1-Rac1-mediated regulation of VE-cadherin has a different effect on established junctions than on those newly formed during brain angiogenesis [65].

## 6. VE-Cadherin Controls S1PR1 Expression through Coordinated Expression of Several Genes to Promote Vascular Stability

All reports in the literature strongly suggest that S1PR1 signaling can regulate VE-cadherin expression and localization to maintain the integrity of the endothelial barrier. We infer the corollary that VE-cadherin could potentially regulate the expression of S1PR1. VE-cadherin has a certain number of cytoplasmic interacting proteins that couple VE-cadherin to the cytoskeleton, which were termed α-, β-, and y-catenin. β-catenin translocates to the nucleus and modulates the expression of several genes through its binding to T-cell factor (TCF)/lymphoid enhancer-binding transcription factors. As an example, β-catenin can activate transcription of Wnt/β-catenin target genes. Since β-catenin does not possess a DNA binding domain, it requires DNA binding partners to bind to the promoters of its target genes [66]. For instance, the zinc finger transcription factor Krüppel-like factor 2 (*Klf2*) was shown to stabilize the vasculature and to limit vascular disease in the face of metabolic and inflammatory challenges. In adults, *Klf2* acts as a central transcriptional switch point between the quiescent and activated states of EC. Indeed, it was shown that *Klf2* overexpression inhibited VEGFR2 promoter activity and, thus, inhibited VEGFR2 mRNA and protein expression [67]. Interestingly, *FoxO1,* which is a direct transcriptional regulator of *Klf2,* regulates T-cell trafficking, quiescence, and survival [68]. The first relation between cell migration and S1PR1/*Klf2* was demonstrated in *Klf2*-deficient T-cells that showed a defect in thymocyte emigration. Because S1PR1 was shown to be critical for thymocyte migration, the authors examined the expression of S1PR1 in *Klf2*^−/−^ thymocytes and found upregulation of S1PR1 mRNA expression. Using chromatin immunoprecipitation (ChIP) assays, the authors demonstrated that *Klf2* both binds and activates the S1PR1 promoter, which is critical for thymocyte egress. During inflammatory processes, the dysfunction of the endothelium is a cause of vascular diseases, and *Klf2* might be interesting as a therapeutic target for the treatment of chronic inflammation [69]. Indeed, the EC detection of a change in shear stress stimulates a signal to the nucleus that results in modification of gene transcription. The expression of *Klf2* has been demonstrated in areas of high shear stress in the adult human aorta [70,71]. The mechanical changes in vessel walls upon shear stress is mediated by the activation of integrins, GPCRs, tyrosine kinases, or ion channels in EC. The best-studied mechanotransducer is a complex of VE-cadherin and VEGFR2 at cell–cell junctions. VEGFR2 activates the PI(3)K-Akt pathway within minutes, leading to the phosphorylation of *FoxO1* and thus facilitating its active nuclear export and inhibiting its transcriptional activity [72,73]. VE-cadherin clustering at junctions in confluent EC was shown to induce persistent *FoxO1* inactivation. As a consequence, nuclear *Klf2* binding to the S1PR1 promoter will be increased, leading to S1PR1 expression and stabilization of the vessel [74]. Significantly, its association with other transcription factors also determines the specificity of the downstream effects of *FoxO1*. For instance, it was shown that β-catenin forms a complex with *FoxO1,* which represses claudin-5 expression [75]. Taken together, these data show that *Klf2* and *FoxO1* may cooperate in the transactivation of target genes and regulate EC proliferation, as well as the vascular barrier.

Forkhead Box (FOX) transcription factor is another transcription factor that has proangiogenic functions. Indeed, its inactivation in EC revealed that *Foxf1* regulates the transcription of genes for angiogenesis and maintenance of EC junctions. Indeed, using chromatin immunoprecipitation and luciferase reporter assays, the authors demonstrated that *Foxf1* directly induced the transcriptional activity of the *S1PR1* promoter, thus increasing the expression of both mRNAs and proteins [76,77]. Moreover, the deletion of *Foxf1* in ECs dramatically reduced the VE-cadherin mRNA, as well as β-catenin. Taken together, these data demonstrate that *Foxf1* is also involved in the stability of the endothelium barrier.

The epigenetic regulation induced by VE-cadherin clustering in human dermal microvascular ECs showed that VE-cadherin is involved in the expression of several genes related to vascular stability [78]. For instance, there was a general downregulation of genes involved in cell proliferation and EC sprouting, compared to a promotion of genes involved in the interaction with the extracellular matrix and cell–cell adhesion. This effect can be attributed to the interaction of VE-cadherin with its intracellular partners, such as β-catenin and p120. Since the truncation of 80 AA in the VE-cadherin cytoplasmic tail prevents the binding of β-catenin [5], the complex formed in the nucleus may increase *FoxO1* activity by stabilizing its binding to DNA. The specificity of the downstream effects of β-catenin depends upon its localization. When at the adherens junctions, β-catenin stabilizes VE-cadherin by anchoring its cytoplasmic tail to the actin cytoskeleton. In response to several stimuli, the dissociation of β-catenin from the adherens junctions acts as a transcriptional coregulator. This implicates phosphorylation processes.

Among the nine tyrosine amino acids in the cytoplasmic tail of VE-cadherin, the phosphorylation of residues Y^658^, Y^685^, and Y^731^ results from activation of kinases and/or suppression of phosphatases. The amino acid tyrosines Y^658^ or Y^731^, in the region of VE-cadherin, were previously shown to bind p120- and β-catenin. With the exception of E-cadherin, the Y^658^ site is well-conserved among cadherins, which suggests that phosphorylation of this residue may be a general mechanism to disrupt cell barrier function. Conversely, since the Y^731^ site is unique to VE-cadherin, we suggest that the regulation of the Y^731^ residue may implicate an endothelium-specific mechanism. We suggest that the uncoupling process might be linked to the reprogramming of ECs through several gene expressions (unpublished observations). Indeed, the tyrosine phosphorylation of VE-cadherin at site Y^731^ reduces β-catenin affinity, leading to an increased amount of cytoplasmic β-catenin molecules available for signaling to the nucleus, and would, therefore, influence gene expressions [79,80]. This includes developmental regulators and other genes involved in coordinating cell proliferation, cell–cell interactions, and cell–matrix interactions. While VE-cadherin has been considered as a multifunctional adhesion receptor modulating endothelial biology, VE-cadherin may also dictate EC behavior through the regulation of intracellular signaling pathways controlling gene transcription via multiple DNA-binding proteins, including, at least, β-catenin.

## 7. Potential Roles for S1PR1 and VE-Cadherin in Therapies

The roles of S1PR1 and VE-cadherin in vascular homeostasis are well-described, but could provide useful targets for therapies to treat diseases associated with endothelial barrier disruption. For example, hepatic ischemia reperfusion injury is induced during the resection and transplantation of the liver. This injury leads to abnormal vascular tone and increases in macrophage and proinflammatory cytokine filtration due to the sinusoidal endothelial cell damage. Administration of S1PR1 agonists leads to an increase in VE-cadherin expression, which attenuates the symptoms of hepatic ischemia reperfusion injury [81].

Ovarian hyperstimulation syndrome is characterized by ovarian enlargement and vascular leakage, and leads to complications with follicular growth and maturation. S1PR1 agonists increase VE-cadherin expression and prevent its phosphorylation. Moreover, a decrease in VEGFR1 and VEGFR2 levels is also observed. In this syndrome, S1PR1 treatment restores the endothelial barrier [82,83].

In the context of cancer, S1PR1 is known to reduce tumor angiogenesis. Evidence for this was derived from two xenograft tumor growth models. The administration of miR302-367 promoted VE-cadherin expression and *Klf2* activity, which, in turn, induced S1PR1 transcription. The results demonstrated that S1PR1 mediated angiogenesis restriction and led to vascular stability. Therefore, S1PR1 has the capacity to reduce tumor growth [84]. In breast cancer, low S1PR1 expression induces the formation of vasculogenic mimicry (VM), which consists of vessel-like networks associated with poor prognosis and the development of tumor metastases. VE-cadherin/Beta-catenin complex is highly expressed in VM. Overexpression of S1PR1 in breast cancer prevents the VM formation by inducing VE-cadherin phosphorylation on tyrosine 731, leading to disruption of the VE-cadherin/β-catenin complex and, thus, a reduction in tumor growth [85].

S1PR1 is also implicated in the immune system and has a significant impact in several diseases. Multiple sclerosis (MS) is an autoimmune disease where the B and T lymphocytes attack the myelin sheath, leading to the nerve conduction deficiency. S1PR1 expressed on B and T lymphocytes is responsible for cell trafficking. Ozanimod is an S1PR1 modulator that is approved for clinical usage in several countries, with the clinical outcomes of reducing inflammation, demyelination associated with MS, and circulating lymphocytes [86]. The blockade of S1PR1 pathways in autoimmune thyroiditis reduces the incidence and the gravity of the disease, along with a reduction in the proportion of inflammation cells [87].

In chronic inflammatory processes such as atherosclerosis, S1P binding to S1PR1 activates the signaling pathway NF-kB, which is responsible for angiogenesis and secretion of proinflammatory cytokines, such as tumor necrosis factor alpha and interleukin 1 beta, from lymphatic endothelial cells. Mice with atherosclerosis that are treated with FTY720, a competitor of S1P, have fewer lymphatic vessels, which implies that S1P/S1PR1 is a pathway involved in vessel tube formation [88].

Altogether, because VE-cadherin is an important member of adherens junctions in all vessels and S1PR1 is mainly expressed in EC, it seems plausible that targeting the cross-talk between signaling pathways might be important to improve therapeutic outcomes.

## 8. Summary

This review describes research in the area of endothelial barrier integrity, which is critical for normal vascular function. We particularly focused on the regulation of endothelial junctions by S1PR1 and VE-cadherin (Figure 2).

Endothelial junctions have a major role in controlling the integrity of the endothelial barrier, and, thus, in maintaining the vascular tone, vascular permeability, and oxidative stress response for healthy blood and lymphatic vessels to avoid vascular diseases. VE-cadherin is the endothelial-specific adhesion protein at the center of this major role. Its potential structural modifications through growth factor pathways can regulate the reorganization of actin, cortactin, and other cytoskeletal proteins, and, thus, have a huge impact on EC junction integrity. Sphingolipids, such as S1P and its receptor S1PR1, have a strong effect on EC and vessel structure. S1P/S1PR1 signaling influences endothelial barrier behavior by regulating cell–matrix and cell–cell adhesion, thus providing a lipid mediator signaling system of fundamental importance in human diseases. We have identified that the Y^731^ site is unique to VE-cadherin, and, thus, we suggest that regulation of the Y^731^ residue may implicate an endothelium-specific mechanism in response to sphingolipid signaling pathways. We suggest that the uncoupling process might be linked to the reprograming of EC through several gene expressions (unpublished observations). This concept is supported by observations that the tyrosine phosphorylation of VE-cadherin at site Y^731^ reduces β-catenin affinity, leading to an increased amount of cytoplasmic β-catenin molecules available for signaling to the nucleus, which, therefore, would influence gene expression [79,80]. Thus, further elucidation of the interplay between VE-cadherin EC junctions and lipid metabolism will be useful for future developmental insights into the search for efficient human therapeutics.

## Figures and Tables

**Figure 1 ijms-24-04018-f001:**
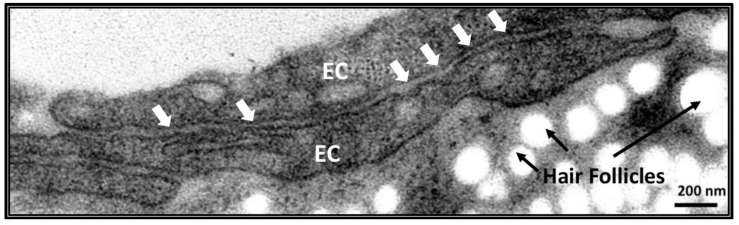
High-resolution transmission electron micrograph showing an endothelial adherens junction in adult mouse skin. EC stands for endothelial cell. The white arrows indicate the adherens junction between two overlapping ECs. The white spots are the hair follicles in the mouse skin. Scale bar is 200 nm.

**Figure 2 ijms-24-04018-f002:**
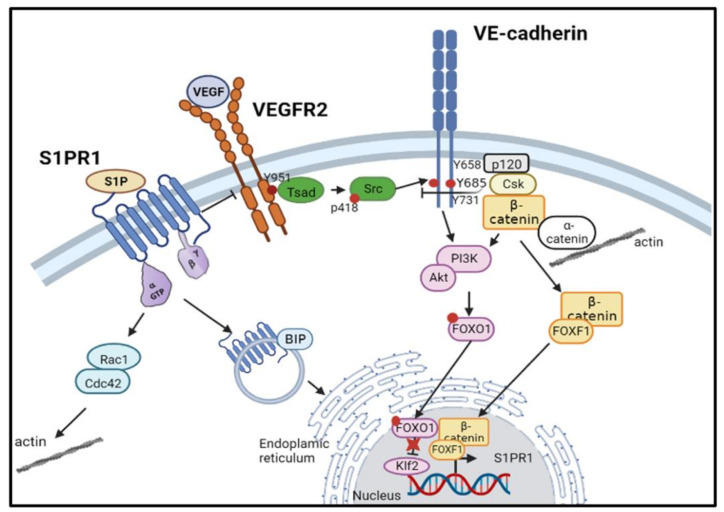
Dialogue between VE-cadherin and sphingosine-1-phosphate receptor 1 (S1PR1) for protecting endothelial functions.

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
