# Peer review of "Dialogue between VE-Cadherin and Sphingosine 1 Phosphate Receptor1 (S1PR1) for Protecting Endothelial Functions"

_ijms, 2023, doi:10.3390/ijms24044018_

Round 1
Reviewer 1 Report
The authors have provided an overview of VE-Cadherin and S1PR1 signaling pathways and how they connect to regulate endothelial barrier properties. The review is concise within the scope of the journal and well-referenced. In my opinion, this review is interesting but does not bring much new ideas to the already extensive available literature. Although the authors have extensively described the molecular mechanisms by which VE-Cadherin and S1PR1 are regulated and can cross-talk, I do suggest as indicated in the abstract that they include an additional chapter on how dysfunctions of adherens junctions in endothelial cells contribute to human diseases in link with S1P. They could also provide some schemes describing the signaling pathways and crosstalk between VE-Cadherin and S1PR1 to help the readers. Finally, there are multiple typos and grammatical errors that make it difficult to always understand what the authors were trying to explain. English language needs to be carefully revised and reviewed.
Author Response
Dear Reviewer,
Thank you very much for the review of our manuscript.
We agree that this review summarizes the existing literature, except for endothelial cell reprograming observed in the Y685 VE-cadherin knock'in mouse and related to S1PR1 that will shortly be published by us. Part of this work was mentioned as unpublished observations.
We appreciate your suggestion to include an additional chapter on diseases. We agree this is an important next step. Please see §7. We hope that this is appropriate.
We appreciate the need to provide a scheme to help the readers. In fact we had included two figures with the initial submission but for some reason you had not received these figures. We have re-included Figures 1 and 2 in the body of the text.
Finally, we apologize for multiple typos and grammatical errors . A native English speaking colleague (Pr D Martin) has corrected the english expression for the revised manuscript.
Reviewer 2 Report
OVERALL
The present manuscript is a review of VE-Cadherin and S1P in vascular integrity and junctional signaling. It is of generally excellent quality and reviews an interesting and relevant body of work. The depth of the review is appropriate, the structure is focused and engaging, and the writing is generally strong.
MAJOR CONCERNS
None
MINOR CONCERNS
- There were more grammatical errors and typos in the manuscript than can be listed here. This manuscript needs to be copy-edited and those errors addressed.
- There is reference to a Figure 2 in the summary that was not included in my version of this manuscript. Rather than to remove this reference, I strongly encourage the authors to include 1 or 2 figures to this manuscript to help illustrate the signaling pathways described in the main text.
Author Response
Dear Reviewer,
Thank you very much for reviewing of our manuscript.
We apologize for multiple typos and grammatical errors. A native English speaking colleague (Pr D Martin) has corrected the english expression for the revised manuscript.
We appreciate the need to provide a scheme to help the readers. In fact it was not included in the main manuscript when it was submitted text so you did not get it. Now we have included Figures 1 and 2 and we hope that this is more appropriateReviewer 3 Report
The manuscript was well-written and I don't have major concerns. It would be nice to include a small paragraph to discuss the role of S1PR1 and VE-Cadherin crosstalk in mediating immune responses.
Author Response
Dear Reviewer,
Thank you very much for reviewing of our manuscript
We appreciate your suggestion to include a small paragraph to discuss the role of S1PR1 and VE-Cadherin crosstalk in mediating immune responses. We agree this is an important next step. A part of the text has been added. See §7. We hope that this is more appropriate with this type of review. Now we have included also Figures 1 and 2 and we hope that this is more clearer.